# Porous Material (Titanium Gas Diffusion Layer) in Proton Exchange Membrane Fuel Cell/Electrolyzer: Fabrication Methods & GeoDict: A Critical Review

**DOI:** 10.3390/ma16134515

**Published:** 2023-06-21

**Authors:** Javid Hussain, Dae-Kyeom Kim, Sangmin Park, Muhammad-Waqas Khalid, Sayed-Sajid Hussain, Bin Lee, Myungsuk Song, Taek-Soo Kim

**Affiliations:** 1Industrial Technology, University of Science and Technology, Daejeon 34113, Republic of Korea; javidmohsin77@kitech.re.kr (J.H.); jhsm8920@kitech.re.kr (S.P.); waqas@kitech.re.kr (M.-W.K.); 2Korea Institute for Rare Metals, Korea Institute of Industrial Technology, Incheon 21999, Republic of Korea; kyeom@kitech.re.kr; 3Chemical Engineering and Applied Chemistry, Chungnam National University, Daejeon 34134, Republic of Korea; sayedsajidh506@gmail.com; 4Department of Advanced Materials Engineering for Information and Electronics, Kyung Hee University, Yongin 17104, Republic of Korea; leebin@khu.ac.kr

**Keywords:** fuel cell, PEM fuel cell, porous material, titanium GDL, fabrication technique, GeoDict

## Abstract

Proton exchange membrane fuel cell (PEMFC) is a renewable energy source rapidly approaching commercial viability. The performance is significantly affected by the transfer of fluid, charges, and heat; gas diffusion layer (GDL) is primarily concerned with the consistent transfer of these components, which are heavily influenced by the material and design. High-efficiency GDL must have excellent thermal conductivity, electrical conductivity, permeability, corrosion resistance, and high mechanical characteristics. The first step in creating a high-performance GDL is selecting the appropriate material. Therefore, titanium is a suitable substitute for steel or carbon due to its high strength-to-weight and superior corrosion resistance. The second crucial parameter is the fabrication method that governs all the properties. This review seeks to comprehend numerous fabrication methods such as tape casting, 3D printing, freeze casting, phase separation technique, and lithography, along with the porosity controller in each process such as partial sintering, input design, ice structure, pore agent, etching time, and mask width. Moreover, other GDL properties are being studied, including microstructure and morphology. In the future, GeoDict simulation is highly recommended for optimizing various GDL properties, as it is frequently used for other porous materials. The approach can save time and energy compared to intensive experimental work.

## 1. Introduction

The major attraction toward renewable energy is due to the limited resources ad pollution of non-renewable energy. Due to the widely increasing demand and attention toward renewable energy such as solar, hydropower, tidal, and wind, there is a need to store the excess energy. So we can store that energy in the form of hydrogen, which can be converted into electricity by using a fuel cell with excellent efficiency, unlike the battery, which has a problem with self-discharge. An electrolyzer can be used to split water into hydrogen and oxygen that can be stored and used as fuel in the fuel cell to reproduce energy. A fuel cell converts chemical energy into electrical energy, and electrochemical reactions occur due to the noble metal catalyst. The produced energy can be used in transport, power production, portable applications, and households, to name a few. The fuel cell’s power ranges from few watts to several kilowatts depending on the application, from small portable devices to heavy buses. In the fuel cell, an electrochemical reaction occurs between the fuel and the oxidant; as a result, water, heat, and electricity are generated. Fuel can be H_2_ or alcohol, to name a few. The hydrogen is diffused into the anode catalyst from the gas diffusion layer where the oxidation reaction occurs, while on the other end, oxygen is reduced with the same pattern [1,2,3]. The cathode and anode reactions are as
Anode: H_2_ → 2H^+^ + 2e^−^Cathode: O_2_ + 4H^+^ + 4e^−^ → 2H_2_O(1)

Additionally, the overall reaction is
2H_2_ + O_2_ → 2H_2_O 

The high-performance fuel cell requires well-designed components. Using the right material and appropriate manufacturing techniques is crucial for making these parts highly efficient. GDL is one of the fundamental parts of the fuel cell. It serves two purposes: first, to transport fluid, heat, and current, and second, to support other fuel cell components. Excellent permeability, better thermal and electrical conductivity, high corrosion resistance, and high mechanical strength are some of the major concerns of GDL. Such a GDL requires a very flexible and controlled fabrication method to generate well-designed pores. Optimization of various properties is also essential for achieving excellent results. Numerous research articles concentrate on GDL parameters, but minimal attention is given to fabrication methods and software optimization. Therefore, this review paper focuses on the various fabrication technologies of porous material, particularly GDL. Each process has a different impact on GDL properties, as the porosity controlling agent is different in each process. Some of the key factors that influence porosity and other features of GDL are demonstrated. In addition, several Geodict simulations are presented to show how this tool can optimize these values with lower energy, cost, and expertise. To sum up, this paper equipped the reader to pick any synthesis routes according to their requirement and limitation without doing a comprehensive literature survey. Choosing a suitable manufacturing method and optimizing various properties using the GeoDict simulation tool can lead a researcher to produce an ideal porous material, especially GDL. GeoDict (Release Version: Service Pack 2) is an advanced multi-scale 3D image processing and material simulation software by Math2Market for characterization, development, and optimization.

## 2. Types of Fuel Cells Based on Electrolyte

The fuel cell can be divided into five basic types based on electrolytes. Some details about these five types are shown in Table 1 [4]. Comparing all these, PEMFC is effective for many applications due to its high-power density, low operating temperature, and quick response [5,6]. 

### Proton Exchange Membrane Fuel Cell

PEMFC can be a future technology due to its higher volumetric power density, high efficiency, lower temperature during operation, and simple, fast startup and shutdown [7]. Fuel cell technology is becoming more popular around the globe due to the rising price of crude oil, global warming, and carbon dioxide gas emissions. One of the factors contributing to its popularity is its declining cost. This price reduction trend can also be seen in the Indian market, as shown in Figure 1a [8]. PEMFC mainly consists of a PEM sandwiched between two catalysts (mainly platinum) layers, anode and cathode, as shown in Figure 1b. The PEM and CL are the heart of the whole cell. Both catalysts are attached to the GDL. These three make a complete membrane electrode assembly, which is the powerhouse of the fuel cell. Each of these three main components plays a vital role in the performance of the whole system. The gas diffusion layer (GDL) assumes the responsibility for fluid distribution, electron conduction, water management, and mechanical support. It ensures the uniform dispersion of reactant gases, facilitates a smooth flow of electrical energy, adeptly balances water levels, and fortifies the structural integrity of the system. On the other hand, the catalyst layer serves as the primary catalyst for electrochemical processes, ensuring proton and electron transport while maximizing surface area to support effective reactions. Meanwhile, the proton exchange membrane (PEM) acts as the guardian of selectivity, enabling exclusive proton conduction while preventing electron and unwanted reactions between oxygen and hydrogen. It ensures the separation of gases, upholding the integrity of the PEMFC. While each component serves a distinct purpose, they converge as an interdependent trio, unified in their mission to propel the PEMFC toward optimal performance. The GDL, catalyst layer, and PEM form an inseparable ensemble, collectively harnessing their individual capabilities to create an extraordinary system of energy conversion. A single MEA has a mean voltage output below 1 V. In order to achieve higher power, we must link multiple MEAs. For this, a bipolar plate is employed.

A bipolar plate generally comprises carbon or other conductive materials to ensure electrical conduction among cells. In addition, these plates also provide mechanical strength to the cell. Finally, the gasket is added to the edge of the MEA to make a gas-tight seal [9]. Therefore, the efficiency of the fuel cell depends on the high-performance GDL and other components. Therefore, due to the critical functional and supporting role of GDL in fuel cells and electrolyzers, understanding the GDL properties, materials, and especially fabrication method is significant.

## 3. Gas Diffusion Layer

GDL is a porous layer of carbon or other conductive material with micrometers thickness. The GDL prevents excess water and gases into the CL and gives mechanical support to the CL. It conducts heat and electrons; hence, it is crucial for the transport and electrolysis of water. It helps the water vapor to reach the PEM and, on the other end, removes the water produced as a byproduct. It also keeps the catalyst layer safe from corrosion and erosion. Without the GDL, fluid pressure would concentrate on specific points, potentially damaging both the PEM and CL. The GDL’s role is to prevent such failures by uniformly distributing fluid and maintaining uniform pressure, ensuring the system operates effectively. The hydrophobic nature of GDL (combined with the hydrophobic agent) avoids pores blockage and makes gas transport easier. There are two types of GDL, one is a single layer (macro porous) which has a thickness range from 100 μm to 500 μm, and the other is a dual layer that contains an additional microporous substrate with a thickness of up to 100 μm. The double layer is employed when there is an intense need to avoid electrolyte flooding. In conclusion, GDL serves two functions: the first is functional, and the second is to support other components [10]. 

Pores can be categorized into micropores (2 nm), mesopores (2–50 nm), and macropores (>50 nm). GDL contains micropores that require high capillary pressure to diffuse water and other gases. Due to this reason, a complete and detailed pores morphology and its uniform distribution should be the main concern for the researcher during the fabrication of GDL, which is the ultimate requirement for the higher efficiency of the fuel cell. The other essential factors that must be considered in the gas diffusion layer are corrosion resistance, diffusivity, permeability, electrical conductivity, thermal conductivity, and mechanical strength. The electrolyzer has high efficiency as compared to the fuel cell. One of the reasons is that in the electrolyzer, gases are a byproduct, which can be removed easily, while in fuel cells, water is the byproduct, and its extraction is harrowing. This property of the electrolyzer hints that if water removal is improved, fuel cell efficiency can be enhanced [2,11]. Some studies have been reported about the effect of a randomly dispersed hydrophilic particle in the GDL for better water management. A significant improvement in the management of fluid flow is the hydrophobic treatment of GDL. Optimizing the hydrophobic treatment of PTFE is necessary to prevent adverse effects on certain GDL features, including thermal conductivity. Achieving a high level of all these qualities requires superior material selection and fabrication with optimum conditions such as thickness and porosity [12]. 

### 3.1. GDL Material

#### 3.1.1. Carbon GDL

The most commonly used GDL is carbon paper and carbon cloth, which have limited mechanical strength at high porosity. When it is subjected to compressive stress, the breakage and displacement of fiber happen; as a result, degradation occurs in almost 50%. In addition, CO_2_ production is chemical degradation which is the reaction of carbon GDL with oxygen. Carbon is not consistent in its properties as its properties vary with the applied pressure [13]. Steel is also a candidate for the GDL, but corrosion is the main weakness. In the highly corrosive environment of the PEM fuel cell, carbon comes out of the steel and causes corrosion, as shown in Figure 2. These failures indirectly reduce the performance and reliability of the PEM fuel cell. To avoid these types of failures, metallic GDL with the optimized parameter will be the best solution. Therefore, researchers are moving towards titanium and its alloys, which needs a proper investigation to find the best operating parameter such as porosity, pore size, and thickness.

#### 3.1.2. Titanium GDL

Titanium is among the top 10 most abundant elements in the world. Some properties of titanium that are our focus for the GDL and their comparison with carbon and steel are shown in Table 2. Titanium has silvery white color and 4.5 g/cm^3^ density, which is almost half as compared to steel with the same amount of strength. It is crucial for the mass control of PEMFC. Titanium and its alloys have broad applications in medical, industrial, aerospace, power production, automobiles, and marine, to name a few. The intense popularity of titanium is its high corrosion resistance, high strength-to-weight ratio, biocompatibility, and low elastic constant. Furthermore, it has a high corrosion resistance because it makes a thin oxide layer of a few nanometers, which acts as a shield against extensive corrosion. As a consequence of this oxide layer, titanium has almost the same corrosion resistance as platinum [14,15]. 

According to Minh Young, the thickness and weight of carbon GDL fabricated by freeze casting are depleted by 6.67% and 3.03%, respectively. In the case of titanium, the observed depletion was only 0.44% in weight, while thickness remained unaffected. In addition, the only problem of titanium GDL is the transfer of fluid, which can be controlled by enhancing open porosity and hydrophobic treatment [18]. Stuart et al. [19] recommended that thin-porous titanium with small and straight pores would be the better selection. 

Carbon, despite its high electrical conductivity, falls short as an ideal choice for GDL due to its low strength and limited corrosion resistance. On the other hand, steel offers high strength but exhibits extremely low electrical conductivity and suboptimal corrosion resistance. In comparison, titanium boasts exceptional attributes, including high strength, lower density, and excellent corrosion resistance. Hence, porous titanium is the best selection for GDL as compared to carbon and steel. However, the challenging aspect is its manufacturing with highly controlled porosity, thickness, and pore morphology since these factors greatly influence the primary properties of GDL. For this purpose, the fabrication methods should be thoroughly examined to make an ideal Ti GDL for PEMFC. 

Porosity is directly related to permeability but is indirect with mechanical, thermal, and electrical properties. Karoenko–Jereba et al. [20] demonstrated increased permeability by increasing porosity. The porosity range is from 78% to 97.5%. However, as a consequence, ohmic resistance also rose to 70 Ω. Wei et al. [21] proposed that the GDL’s water transport might be enhanced by keeping porosity lower near the CL and higher towards the flow field. Powder metallurgy and fabrication of porous metallic GDLs with graded porosity can be used to investigate this further. 

GDL thickness is another essential component that affects GDL performance. It should be considered when designing GDL because it has a direct relationship with mechanical properties and an inverse permeability [22]. Stuart et al. [19] evaluated three thicknesses (170 μm, 278 μm, and 534 μm) and found that as the thickness of the GDL reduced, the Ohmic loss and transport losses dropped, which led to an improvement in performance. Thickness has a significant impact when the variation is large. 

Thermal conductivity considerably influences the GDL performance, as it keeps temperature and current density uniform. High thermal conductivity helps remove water in the vapor form while keeping the cell cool. The optimized range of thermal conductivity is from 0.2 to 0.6 Wm^−1^K^−1^. Similarly, the electrical conductivity of GDL is essential because it serves as a conduction bridge between the CL and the bipolar plate. When the porosity increases, permeability rises, but conductivity and mechanical characteristics decline. Therefore, optimizing porosity is necessary for an efficient system. One study revealed that the highest power generation of the cell is at 60% porosity, as shown in Figure 3 [12,22,23,24].

In addition to porosity and thickness, pores composition (size, shape, and distribution, to name a few) is the important feature that decides all the properties of porous material. For instance, open pores and uniform distribution can improve the electrical conductivity, permeability, and strength of the GDL. Pores can be varied and controlled by various fabricating methods as the porosity-controlling agent differs in each method [25,26,27]. The diffusion phenomenon is very complex in the PEMFC due to the solid phase of GDL, liquid water, and (oxygen and hydrogen) gases. It becomes more complex during the operation due to the applied load. We must research the pores and how they relate to these features to obtain the best combination of them at once. To understand all these parameters and their controlling agent, we decide to make a review article focusing on the fabrication methods. This will felicitate the readers to choose one of these methods according to their requirements [28,29,30].

## 4. Fabrication Technologies

Nowadays, porous material has received incredible amounts of attention due to its vast application in industry, wear-resistant tools, heat exchangers, biomedical and other applications. In addition, it can be used in the separator for batteries and other energy-storing devices, where we have to deal with the mass, heat, and charge transfer. The final characteristics of the porous material depend on the porosity, thickness, and pore morphology, which determines the application of porous material. Among all the parameters, the fabrication technique is dominant in affecting the properties of the porous material. Especially when the porosity is controlled with different porosity controller agents such as in tape casting porosity controller is sintering temperature. 

In the phase separation technique, porosity is controlled by the pore agent. In additive manufacturing, input design is the controlling parameter, ice control in the freeze casting, and finally, etching time and mask width govern the porosity in photolithography [31]. This review is about some of the crucial methods widely growing in the fabrication of porous material, as shown in Figure 4.

### 4.1. Tape Casting

Tape casting is a very commercial and inexpensive method for the fabrication of porous material. Tape casting has high control on thickness up to micro size. Since high-performance GDL requires a shallow thickness, therefore, by employing tape casting, we can easily fabricate our desired GDL. Tape casting is composed of slurry formation, casting, debinding, and sintering. Each has its own importance in the fabrication of porous tape. The slurry and green tape fabrication with the overall casting setup are shown in Figure 5a–c. Making slurry with the optimum condition is the first step of tape casting, which includes mixing the powder with numerous additives to improve the rheological properties of the slurry. The slurry is separated into two types on the bases of solvent: aqueous and non-aqueous.

Every solvent used for tape casting should be non-toxic, cheap, and readily available. Therefore, water is highly preferred, but sometimes it has limitations. Slurry formation in two or three steps is preferred for better rheological properties and improved additive solubility. Furthermore, polar solvents have a high viscosity as compared to non-polar ones. Faster drying solvents improve overall efficiency. However, rapid tape drying might result in cracks or another failure [32].

A dispersant is added to the slurry to reduce the viscosity of the slurry and eliminate particle attraction. Dispersant allows lower solvent, resulting in faster drying and enhancing debinding efficiency. It can be optimized by rheological and sedimentation testing [33]. Zhang et al. [34] compared five dispersants and reported that Phosphate ester showed the best performance by showing lower sedimentation rate and viscosity. They concluded that the best dispersant for TiO_2_ slurry is phosphate ester with a binder. On the other, if there is no binder, then side polyvinyl butyral is the appropriate selection. A Binder is the fundamental component of the mixture since it connects particles and provides strength and flexibility to the green tape. If we add binder first in the mixture (solvent and powder), it will attach several particles, resulting in a reduced density of green tape. The plasticizer is used in the tape casting to attain the plasticity and flexibility of the green tape. It changes the transition temperature of additives in the debinding process. Plasticizers can be internal or external. The internal is involved in the reaction, while the external plasticizer is solely concerned with lubrication [34,35].

Moreover, the evaporation rate is also considerable in tape casting, as cracks arise in the green tape when the evaporation rate is not optimized [36]. Every part has its own effect on the slurry. Hence, the optimization of every additive (solvent, binder, dispersant, plasticizer, and surfactant, to name a few) is required for better slurry and tape. Some of the additives are shown in Table 3.

The powder size and its distribution affect the density of the green tape. For instance, Rauscher et al. [33] reported that fine powder’s viscosity is higher than coarse powder, which needs more dispersion. They used 1% dispersant for fine and 1.25% for coarse powder with 57% and 56% density, respectively. In addition, the spherical powder is better for the stability of the slurry [36]. Anchalee Manonukul et al. [37] used 1.5% PVA as a binder, 1% Dolapix as a dispersant, and 22.5% water to make a slurry of 75% titanium powder. He reported that the viscosity of titanium slurry increases when we add solid loading or PVA, while dispersant has almost no effect. JE Bidaux et al. [32] made titanium hydride slurry by using titanium powder, methyl ethyl ketone as a solvent, ethyl methacrylate as a binder, and dibutyl phthalate as a dispersant with a composition of 75%, 16.7%, 6.5%, and 1.8%, respectively. P.LI et al. [38] made Ti64 slurry having 75 wt% powder (10–80 μm), 18.0% demi water, 1% Dolapix (dispersing agent), and 0.5% methylcellulose was added as a binder. Other ingredients were added in small amounts for better rheological properties. Polymeric sponges were used as a pore agent to obtain high porous (90%) Ti64, as shown in Figure 5d [38].

E Bidaux et al. [32] performed sintering under 1000 °C and achieved the plate with 25% porosity and 150–320 Mpa strength. S.ahmad attained lower density while sintering at a higher temperature [39]. NF. Daudt et al. [40] fabricated titanium GDL and showed a relation between the porosity and sintering temperature where the sintering range was 1100 °C to 1300 °C and porosity was 7.6–10.5%. Similarly, Franz et al. [41] represent the dependency of the porosity of GDL on sintering temperature that decreases from 55.6% to 6.6% by 800 °C to 1200 °C, respectively. All these demonstrate that the primary factor influencing tape-casting porosity is sintering temperature. Porosity can be increased by using some pore agent such as polyethylene or any other agent we will discuss in detail in the (Phase Separation Technique) PST. For instance, J.P. Li et al. [14] use polyethylene (35–45 pores/inch) to make highly porous titanium for medical uses. Some properties of titanium (and its alloy) GDL manufactured via tape casting along with their porous structure are shown in Table 4.

**Figure 5 materials-16-04515-f005:**
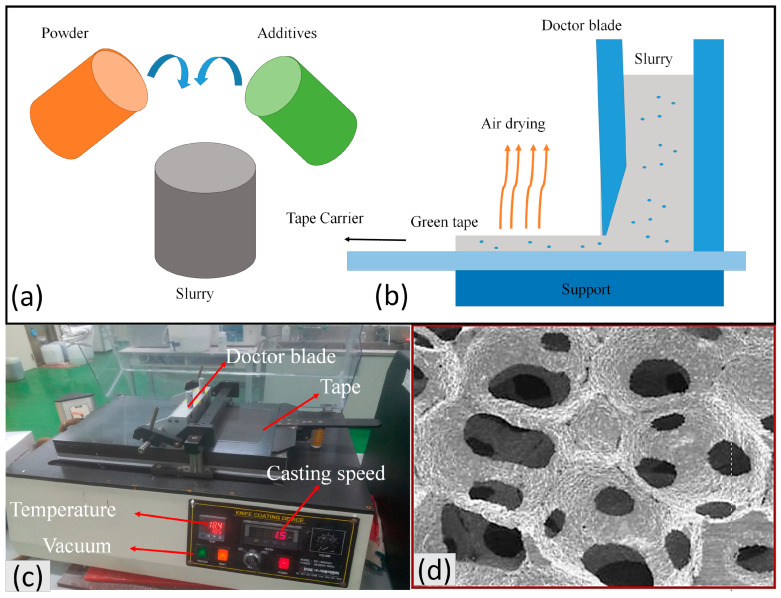
(**a**) The mixing of powder and additives to make the slurry. (**b**) Fabrication of thin green tape from the slurry, (**c**) experimental setup and various controlling parameters of tape casting, and (**d**) SEM image of 90% porous Ti64. Reproduced with permission from [38]. Copyrights 2004 Springer.

### 4.2. Additive Manufacturing (3D Printing)

Nowadays, additive manufacturing is the most powerful technique for fabricating porous material with complex internal and overall designs having enhanced properties compared to the other fabrication methods. The Additive manufacturing technique is well known for its versatility and accuracy [43]. 3D printing is not just restricted to the porous material. However, it has been widely used in the biomedical, transportation, health sector, aerospace, marine, power sector, fashion industry, firearm industry, education sector, food industry, culture heritage, and museum-based digital twin, to name a few. As shown in Figure 6a. the 3D printing (rapid prototype) includes making a design (by 3D scanning or cad designing or photo scanning), then it is exported in an STL file, followed by the G-code generation for slicing the model and then printing by a 3D printer which moves in x, y and z direction to print layer by layer. Finally, the surface is machined if it is needed. In metallic printing, the powder is fused into the designed pattern layer by layer with the help of high-density radiation with high accuracy [44,45].

Powder for metallic printing should be in the size of a few micrometers and spherical for complete packing and a highly dense printed model without any cracks. Conventionally, 3D printing was limited to a few materials, but now it can print polymer, plastics, metal, ceramic, and glass, to name a few. Seven methods of 3D printing are currently used for printing these materials, as shown in Figure 6b. Every method has its pros and cons; therefore, an appropriate manufacturing method should be chosen before printing [43].

For titanium GDL, powder bed fusion is mainly preferred. Many researchers used sintering laser melting and electron beam precisely because of their high power-density electron and laser beams. Three crucial parameters in these types are beam power, beam diameter, and scan speed [46]. Jingke et al. [45] employed plasma-atomized powder of 45–105 μm and electron beam melting 3D printing to create well-controlled, precise titanium GDL with lower thickness and increased porosity. About 100–150 μm spot size was employed in his experiment. 3D printing is unique due to its uniform distribution and complex design of pores. Arunkumar Jayakumar et al. [48] used selective laser sintering to fabricate GDL. Aluminide was chosen as a base material due to its high strength and thermal load. Titanium (70%, 80%, and 90%) was added as a functional material. Enhancing GDL’s transport properties increases the performance of fuel cells and electrolyzers. Therefore, we might maximize the cell’s overall performance if we design the GDL microstructure in an organized manner and print accurately. Daniel Niblett et al. [49] reported that the control design of the microstructure of GDL could significantly impact the fuel cell and electrolyzer performance. When traditional GDL was substituted with 3D printed GDL, a reduction of 0.12 Ωm^−1^ was observed. This happens because 3D printing allows much control over the microstructure of porous materials. The 3D-printed pattern verifies the uniform distribution of pores, which ensures uniform current and heat distribution. Daniel Niblett et al. [49] used three software (AutoCAD, Autodesk, and MATLAB) to simulate the conductivity and permeability of GDL. They compared helix and cubic lattice structures with GDL of no regular pattern. They demonstrated that GDL with properly engineered microstructure has superior results [45,48,49,50]. Xuan Pei et al. [47] reported the difference between the 3D measured and the calculated porosity of porous titanium, as shown in Figure 6c. The difference in porosity is 6.3% when the strut diameter is lower and 0.5% at a higher diameter. This indicates that managing several variables can give 3D printing more precision. Shokou et al. [46] showed various metallic 3D printed porous products with complex geometry, which proves the flexibility of 3D printing. 3D printing can design and print complex geometry GDL with the optimized condition. However, additional enhancements are still required to provide greater control over the accuracy, printing time, and cost.

### 4.3. Freeze Casting

Freeze casting (also called ice templating) is an amazing process for producing porous material with aligned pores that are adaptable to various materials. This aligned porous material made by freeze casting has wide applications, especially in the biomedical industry. Freeze casting involves making slurry and pouring it into the mold, followed by controlled solidification and sublimation. Finally, the debinding and sintering processes are carried out to remove the additives and increase the strength of the porous material, respectively, as shown in Figure 7 [51].

As we discussed in the tape casting, making a slurry with optimized properties is the initial and vital step of freeze casting. The slurry consists of powder (ceramic or metal) and additives. Additives are employed to improve the slurry’s rheological characteristics. The metal or ceramic particles are sustained in the suspended form in the slurry. Slurries are classified as aqueous or non-aqueous based on their solvent. Mostly used non-aqueous solvents are camphene, camphor–naphthalene, and tert–butyl alcohol having different crystalline morphology that is cellular, dendritic, and prismatic, respectively. In contrast, water has lamellar crystalline morphology. The aqueous slurry is mainly preferred because it is non-toxic, readily accessible, and flexible to many materials. The additives used in the slurry play a critical role in tailoring the pores morphology. For instance, the particle can be distributed uniformly by dispersant, which results in the uniform distribution of pores.

Moreover, the viscosity, dispersion of particles, and freezing parameters can be supervised by additives. The particle size and the powder density also have a significant effect on the final product. To sum up, the optimization of slurry is crucial for achieving the desired structure of GDL or other porous material [52]. 

In freeze casting, solidification is the controlled process that decides the pore morphology. The solidification can be unidirectional or multi-directional, as shown in Figure 4b. The solidification process depends on the temperature, power source, rate, and direction. The pore morphology is affected by a variety of factors. For example, if the solidification velocity is high, narrow and thinner pore channels will form. Solidification is the crucial phase in freeze casting because it determines the crystal morphology. This solidification process has been the subject of numerous investigations. The thermal conductivity of the powder and the solidification front are both essential. If the particle has high thermal conductivity, the interface will be concave, and vice versa. The suspension can be made at room temperature and frozen to optimal condition. The pores can be tailored through freezing conditions such as freezing temperature, freezing rate, and time.

Furthermore, directional freezing is favored for complex geometry. After solidification, the solvent is extracted from the suspension through sublimation which is the direct conversion of the solid into a gaseous phase under the controlled condition of pressure and temperature. Pores are generated in the material after the removal of the solvent with the same monolith as that of ice. The sublimation did not widely affect the microstructure of the porous material in freeze casting. Various types of freeze casting and the alignment of pores are shown in Figure 8 [52,53].

HHyelim Choi et al. [54] reported the fabrication of GDL through freeze casting. The first slurry was made by adding 0.28 g PVA and 10 mL of distilled water to 11.25 g of Ti powder. Then, the sublimation was done at −90 °C and 5 × 10^−3^ torr pressure for 20 h. Finally, de-binding was processed at 300 °C for 3 h and sintering at 1100 °C for 7 h. It was shown that GDL made via freeze casting has a current density of 462 mA cm^−2^, which was almost 166% higher than conventional carbon GDL.

Min Hyoung Kim et al. [55] compared titanium foam created by freeze casting and conventional GDL and found that the freeze-casted GDL has open and three-dimensional pores, as shown in Figure 9a, b.

To summarize, solvent, additives, solidification, sublimation, and sintering condition each affect the pores’ morphology. Therefore, each should be appropriately optimized before utilizing freeze casting. Table 5 shows some of the factors that influence the pores’ morphology.

M.B. Frank et al. [61] used three powder sizes of surface magnetized alumina, and three magnetic fields were applied transverse to the ice growth. The 350 nm and 75 mT combination offers the highest horizontal lamellar wall alignment and better strength. The alignment of the pores depends on the temperature, slurry viscosity, and particle size, to name a few. It was suggested that if these parameters are thoroughly investigated, the final properties can be increased even further.

Freeze casting is among the prestigious techniques for fabricating porous material (GDL) with aligned pores. Some reviews about freeze-casting discuss the basic principle, recent progress, various material, and the application of that freeze-casting fabricated material, to name a few. However, our concern in this short review is to connect the freeze casting with other fabrication methods of GDL.

### 4.4. Phase Separation Technique (PST)

Nowadays, porous materials are widely used, and their application is determined by whether the pores are closed or open. Porous materials with closed pores are used in structural applications, while porous materials with open pores are concerned with functional applications. Open porous materials are preferred when high permeability is required, such as in GDL [32]. The Phase separation technique is a versatile technique for the fabrication of porous material. In this technique, pore agents play a key role in regulating porosity and pore morphology (pore size, shape, and distribution). The porosity and its composition can be easily altered by varying the characteristics and percentage of the pore agent. The pore’s size and quantity depend entirely on the pore agent [62].

In PST, there are two sorts of pores: inherent and generated. The inherent pores are obtained by the powder’s size, shape, and distribution, whereas the generated pores are created by removing the binder and pore agent in the de-binding process [63]. Therefore, the steps included in the PST are mixing powder, pore agent, and binder, then compression of the mixture and removal (de-binding) of the pore agent and binder by various means (dissolving, melting, and thermally decomposing). Finally, the sintering process improves the porous material’s mechanical properties, as shown in Figure 10a. As mentioned above, the extraction of pore agents in PST can be accomplished through solvent dissolving, melting, or thermal decomposition, which is why it is segmented into three subtypes [31].

#### 4.4.1. Sintering Dissolution Process

In this process, the pore agents must be soluble, non-toxic, low cost, and have a high melting point, as low melting pore agents can be easily removed with the other two methods. In this process, the pore agent dissolves in the water (solvent) before sintering and comes out of the material, generating pores. In addition, the sintering variable also has a massive effect on the porosity and mechanical properties of the product [64]. 

The steps included in the sintering dissolution process are mixing, compaction, dissolving, and sintering, as shown in Figure 10b. Some of the studies have been reported about the sintering dissolution method using various pore agents in Table 6.

#### 4.4.2. Thermally Stimulated Decomposition

This method is the same as the sintering dissolution method, except the pore extraction is done with the melting process instead of the solvent dissolution process. In this process, low melting, non-toxic, and non-contaminating pore agents are preferred to be easily removed at low sintering temperatures. In addition, the complete extraction of the agent will reduce the contamination. 

The sintering parameters should be focused on carefully to avoid any failure (cracks, contamination, to name a few) in the fabrication of porous material. When the pore agent or the material is reactive at a higher temperature, vacuum sintering is employed to avoid impurities. Table 7 lists some studies [5,6,65]. 

#### 4.4.3. Thermally Melted Elimination Technique

This is a prestigious method for fabricating porous material. In this method, the pore agent is extracted by melting in the sintering process rather than decomposition and dissolving. The whole process is the same as the thermal decomposition, as shown in Figure 10b, except for removing the pore agent by melting during the sintering process [18,43]. The two studies of Mg and ice as pore agents are shown in Table 8.

These three methods are almost the same; the sole variation among these approaches is removing an agent by a different process. However, as the pattern of pore agent extraction differs, various pore agents are employed in the PST, as shown in Table 9. 

The dependency of physical qualities on the shape of the pores is visible, as the spherical, angular, and needle-like pores in titanium have a compressive strength of 185, 175, and 140 Mpa, respectively. The pores are formed as a result of extracted pores agents; therefore, the morphology and quantity of pore agents indirectly impact the physical properties of porous material. The PST approach can produce spherical, interconnected pores with a wide range of sizes [1]. 

In addition to Table 9, Tomoyuki Fujii et al. [66] also fabricated the porous titanium using NaCl as a space holder and spark sintering. The Porosity and pores size ranges were 26–80% and 75–475 μm, respectively. PST is an easy and commercial method for making porous material; however, making lower-thickness material is very challenging. For this purpose, an improvement is needed in the PST. For instance, Van–Tien Bui et al. [67] utilized an improved separation technique to make a very-low-thickness porous polymer film of 1.6 μm. The composition of the mixture controlled the pore size and porosity. The pores size was from 0.26 μm to 1.1 μm. Moreover, Zhongchen He et al. [68] used the induced phase separation method to manufacture polyphenylene sulfide porous membranes while using polyethersulfone as a pore agent. The porosity range was 33–55%, and the average pore diameter was from 23 nm to 85 nm. The thickness of the membrane was kept at 400 μm. The whole PST process is composed of the mixing and complying additives with the powder, followed by the removal of additives and the sintering process. Each stage must be controlled and optimized for the ideal porosity, thickness, and pore morphology. As these parameters mostly decide the final properties of the porous material. PST is usually a controlled process, but due to the inadequate removal of the pore agent and other additives, the final product has some contamination which causes the degradation of the final properties. Further improvement is recommended for PST for making GDL with lower thickness, minimum contamination, higher porosity, and well design pores [69]. 

The implementation of an embedded cenosphere technique presents an intriguing possibility for porous material fabrication. However, it is important to note that this method tends to result in closed pores, limiting its suitability primarily to structural applications. In contrast, when considering applications where high permeability is a paramount requirement, such as in gas diffusion layers (GDLs), the embedded cenosphere technique falls short. GDLs necessitate exceptional permeability to facilitate efficient fluid transport and enhance overall performance. Therefore, alternative methods that prioritize achieving the desired high permeability are more suitable for the production of GDLs [31].

### 4.5. Lithography

A well-designed GDL can increase the diffusivity of the reactant and product fluid to obtain high performance of the cell. In order to achieve this, lithography is also one of the important techniques for fabricating porous GDL. Lithography is micro and nanotechnology, which can be classified based on exposure to radiation into the following subtypes.

Electron beam lithographyIon beam lithographyPhotolithographyX-ray lithography

Photolithography or optical lithography is a flexible technique with accuracy from micrometers to nanometers to create porous material at a low cost. In this method, the pattern made on the mask is accurately transferred to the substrate or underlying metal (thin titanium plate) with the help of light exposure and etching, specifically wet etching. Consequently, the thin plate is converted into a porous plate with the exact pattern as guided by the mask [2,70]. The essential components of photolithography are photoresists, photo masks, and light sources, as illustrated in Figure 11a. Positive and negative photoresists are two types of photoresists. The exposed area to light is removed in positive, while negative photoresist has the inverse mechanism. For making uniform porous GDL, pores morphology on the mask should be accurately distributed. The exposure of the material to light and the process (dry or wet) are the two most important phases in this technology; these two processes control the whole design of the porous material [71,72].

Feng–Yuan et al. [2] demonstrated the relation among the mask size, time, and pore ratio. At the same parameters, porosity was seen from 21% to 35% when the time increased from 20 to 30 min, respectively. Similarly, porosity raised about 11% and 15% with the variation in the pore distribution, as illustrated in Figure 11b. This shows how these parameters affect and govern porosity. Furthermore, the microporous layer with a pore diameter ranging from 100 nm to 1 µm was made. The differences in the mask and final pores (elliptical) are shown in Figure 11c. Kazuyoshi et al. [73] also created 10^−6^ m and 10^−5^ m Metallic GDLs, claiming that by maintaining the porosity constant, the thinner and smaller through-hole diameters performed well [2]. 

Sangwon et al. [70] use photolithography to create micro robots with porous bio-scaffold. Two beams were employed to make a single ellipsoidal spot as the building block. The pore size ranges for the designed and measured samples were 13.22 μm–24 μm and 10.2 μm–21 μm, respectively [74]. From here, we can conclude that by employing this sophisticated method, it becomes possible to achieve the desired pore size range essential for optimal GDL performance. This underscores the remarkable capabilities of lithography in tailoring porous structures to meet the significant demands of GDLs in various applications. Elli Kapyla et al. [75] employed two-photon polymerization lithographic technique with 100 mW power and 13 kHz frequency to make a porous scaffold. The simulated and experimental porosity of the scaffold were compared, and almost 13% deviation was found due to the shrinkage factor. 

The application in an insulator, catalyst filter, lightweight structural material, and sorption media makes the porous material popular around the globe. The lithographic method can be used to create a variety of porous materials, including GDL. Recent advancements have expanded the flexibility of this technique. For instance, the advanced lithographic technology known as direct laser interference lithography is mask-free, easily adjustable, controllable, and cheap. This technique uses numerous beams that are interfered with to increase effectiveness. This approach has seen considerable modifications. However, there are still specific challenges, such as controlling the 3D structure of pores [76].

These five methods are widely utilized for producing porous material, specifically GDL. However, each has unique benefits and limitations. A small comparison is shown in Table 10. The performance of a gas diffusion layer (GDL) in a proton exchange membrane fuel cell (PEMFC) is closely linked to the fabrication method employed. 

Tape casting, a widely used method, allows for the precise control of GDL thickness and uniformity. This can significantly impact the overall performance of the PEMFC, as an optimized thickness ensures efficient reactant transport and minimizes electrical resistance. Additionally, tape casting allows for the integration of additives to enhance properties such as hydrophobicity or corrosion resistance, further influencing GDL performance. Freeze casting, on the other hand, offers the advantage of creating well-defined, interconnected pore structures. By manipulating the freezing conditions, it is possible to control the pore size, morphology, and distribution within the GDL. This plays a crucial role in facilitating fluid transport, optimizing reactant distribution, and minimizing mass transfer limitations, ultimately improving the overall performance of the PEMFC. 3D printing provides a versatile approach to fabricating GDLs with complex geometries and customizable designs. By precisely controlling the printing parameters, such as layer thickness and pattern, it is possible to tailor the GDL architecture to enhance reactant distribution and optimize the electrochemical reactions within the fuel cell. This can lead to improved power output and overall performance.

The GDL produced by PST can exhibit a controlled pore size and morphology, directly influencing its permeability and fluid distribution capabilities. Lithography techniques enable the fabrication of GDLs with precisely controlled pore sizes, shapes, and arrangements, promoting reactant distribution and enhancing the interfacial contact between the GDL, catalyst layer, and electrolyte. This can lead to improved reaction kinetics, reduced mass transport limitations, and enhanced fuel cell performance.

In summary, the choice of a fabrication method for GDLs significantly impacts their performance in PEMFCs. Each technique offers unique advantages and controls over parameters such as pore structure, thickness, and surface morphology. By carefully selecting and optimizing the fabrication method, it is possible to tailor the GDL properties to meet the specific requirements of the fuel cell, ultimately enhancing its efficiency, power output, and overall performance.

In addition to these methods, some other methods, such as the sol-gel or Porosfier method, can be employed for manufacturing GDL as presented by Arnout Imhof et al. [74] utilized the sol-gel method for making porous titanium foam with spherical pores. The calcination temperature was maintained between 600 °C and 1000 °C, while the porosity was kept around 93%. 

To summarize, there are numerous techniques apart from these that can be utilized for the creation of porous material. However, we still need a more advanced method with all the necessary capabilities, such as good control over thickness, porosity, contamination, complex pore design, uniform pores distribution, pores connectivity, and flexibility in pore size and alignment, to name a few. These enhancements will result in an ideal porous material employed for either supporting or functional purposes. A material with high permeability can be created without significantly affecting its mechanical, thermal, and electrical properties. One recommendation is to combine two or more methods to control the parameters as effectively as possible. For instance, using a pore agent in the tape-casting process can produce highly porous and lower-thickness material. Adding or removing some processes in these methods will give us a more versatile technique. Similarly, combining simulation with these techniques will simultaneously lead to an easy optimization of multiple parameters.

## 5. GeoDict

The PEMFC converts chemical energy to electrical energy. However, the operation in the PEMFC is very complex due to the various phases, porous structure of GDL, and the transport of mass, heat, and charge simultaneously. For high performance, GDL must possess high conductivity, good strength, excellent corrosion resistance, and high permeability. For this purpose, the porosity, thickness, and internal structure of the GDL should be thoroughly investigated. However, experimentally, it is challenging, time-consuming, and expensive [77]. Therefore, software modeling and simulation is the ideal approach for these complex problems. Currently, the majority of research on GDL is experimental, with limited attention toward simulated work, and there is almost zero research on the prediction of properties of porous titanium GDL manufactured by powder metallurgy. In the future, software prediction will play a crucial role in the upgraded design of GDL and the optimization of its numerous properties. 

One widely used simulation method is computational fluid dynamics (CFD), which employs numerical algorithms to solve fluid flow equations within the GDL structure. CFD simulations provide insights into the macroscopic behavior of fluid flow, species transport, and heat transfer. They are effective in capturing the overall performance of GDLs in terms of pressure drop, mass transport, and reactant distribution. CFD has limitations in capturing detailed microstructural features and atomistic-level phenomena in GDL materials.

Finite element analysis (FEA) is another commonly employed method for GDL simulations. FEA focuses only on mechanical aspects, such as stress and strain distributions within the GDL. Molecular dynamics (MD) simulations, on the other hand, offer atomistic-level insights into the behavior of GDL materials. They provide information about the interactions between individual atoms, molecular diffusion, and chemical reactions within the GDL structure. MD simulations are particularly useful for investigating fundamental processes, such as the adsorption and desorption of species, at the nanoscale.

Geodict, specifically designed for porous media simulations, combines various numerical methods to provide a comprehensive understanding of GDL behavior. It offers advanced capabilities for modeling complex microstructures, such as the realistic representation of pore networks and tortuosity. Geodict’s unique features include the ability to generate realistic 3D micros size geometries, simulate fluid flow and transport phenomena, and analyze the mechanical behavior of GDL materials. Its unique capabilities for porous media simulations make it well-suited for studying complex transport phenomena and optimizing the performance of GDLs in PEMFCs. It enables researchers to explore the impact of various geometric and material parameters on GDL performance. 

The GeoDict is the most comprehensive approach for multi-scale 3D image processing, material modeling, visualization, material characteristic analysis, simulation-based material development, and process optimization. This software offers a comprehensive quantitative and qualitative analysis of the physical and geometric structure and its effects on the properties of the final product. It allows the characterization of conventional material and the examination of a wide range of novel structural alterations. Furthermore, it is flexible to simulate a variety of porous material features rather than being restricted to one. Some of the modules that are concerned with some specific properties have been shown in Figure 12a [78].

This section of the review aims to provide a detailed analysis of GeoDict and demonstrate how many researchers have utilized and validated this software to analyze various properties of porous material, mostly created from fiber. They investigate permeability, and mechanical, electrical, and thermal properties, to name a few. They also depend on porosity, thickness, fiber diameter, and orientation. The same pattern can be followed for titanium GDL fabricated by the powder metallurgy.

Willi Pabsta et al. [79] analyzed the young modulus and thermal conductivity. GeoDict was employed to simulate several predictions with spheroidal and spherical pores of three aspect ratios, and a good approximation was found. Tereza Uhlířová et al. [80] also predicted the same properties; however, the driving parameters were concave and convex pores. Jaroslaw Sar et al. [81] computed several parameters of the porous material, such as porosity, surface area, percolation path, and tortuosity factor, using the PoroDict module. They determined the fraction of open, closed, small, and large pores. This simulation and prediction were for CGO-LSCF composite oxygen electrodes, which resemble our porous GDL in specific ways. Pabst et al. [82] predicted different results for the young modulus, bulk modulus, and shear modulus, along with thermal and electrical conductivity, they found how porosity impacted these attributes, and a significant impact was seen. They simulated the partially sintered ceramic. All these properties are the same for titanium GDL as well. Therefore, before going to any fabrication method, the necessary parameters should be studied by GeoDict. Youssef Alilou et al. [83] utilized GeoDict and Ansys to determine the flow in high-efficiency particulate air filter pleats and compared the simulated data of this software with the experimental data. Both software can estimate the flow velocities and other flow properties.

For a very accurate result, a fine voxel of 5 μm is necessary for GeoDict. Andreas Grießer et al. [84] also recommended software used in the industrial world to optimize various parameters without creating foam experimentally [83]. H bai et al. [85] employed FilterDict to simulate effective diffusion coefficient and filtration performance w.r.t particle diameter and reported about 0.6% deviation from the experimental result, confirming the high accuracy of the GeoDict. Additionally, an increment in tortuosity was noticed along with fiber diameter. Ortega et al. [86] investigated the morphological and transport characteristics of sand. The permeability was measured experimentally and compared with the simulated results. The values of simulated and experimental permeability were 2.5 × 10^−11^ m^2^ and 2.02 × 10^−11^ m^2^, respectively. A negligible difference was noticed. Furthermore, Yunze hui et al. [87] successfully characterize the 3D model of a plastic layer using GeoDict. It was claimed that the GeoDict 3D model gives more comprehensive information on the structure and properties of our model. Dennis Hoch et al. [88] compared GEODICT and StarCCM+ with the experimental result displayed in Table 11. It can be seen that the accuracy of StarCCM+ is a little high, but the simulation time is extremely long and inconvenient compared to GeoDict.

**Table 11 materials-16-04515-t011:** The efficiency of the clean filter section was predicted by the simulation and obtained experimentally [88].

	Filter Section	Upscaled for the Whole Filter	Experiment	Simulation Time
GeoDict efficiency	2.6%	96.96%	96.5	12.88 weeks
StarCCM+ efficiency	2.8%	97.6%	96.5	10 min

Xiaoguang et al. [89] analyzed the mechanical properties of rock wool board (for thermal insulation) by GeoDict. They used fibers with a diameter of 10.5 μm, 12 μm, and 9 μm. A 5% variation was noted when the comparison was made with the experiment data. They reported that when fiber diameter increased, fiber number reduced, and pore size increased. He Bai et al. [90] also achieve the same trend of growing pore size by increasing the fiber diameter. M. Amin et al. [91] predicted that the fiber tows would be permeable. When comparing two different prediction methods, GeoDict and COMSOL, it was found that GeoDict had a high degree of accuracy. Stefan et al. [92] used DiffuDict for the simulation of the diffusivity of the filamentous fungal pellets. The diameter of the pallet was kept in the range of 410 μm–570 μm.

Additionally, they investigated the link between diffusivity and hyphal fraction. Zhengyuan et al. [93] made a comparison of simulated average pore size, permeability, and collection efficiency with the experimental data. The performance of Geodict was excellent between 5.54 μm and 12.4 μm; however, it was less efficient out of this range, as mentioned in Figure 12b. Sudhakar et al. [94] simulated water permeability for porous media and predicted the result of three types of arrangement of fiber, as shown in Figure 12c. Additionally, the in-plane and through-plane values were measured. A deviation of 0.07 was seen between the experimental dimensionless effective permeability of 2.19 and the simulated dimensionless effective permeability of 2.12. It was recommended that the highly accurate result of GeoDict needs an input of high-resolution microstructural data.

**Figure 12 materials-16-04515-f012:**
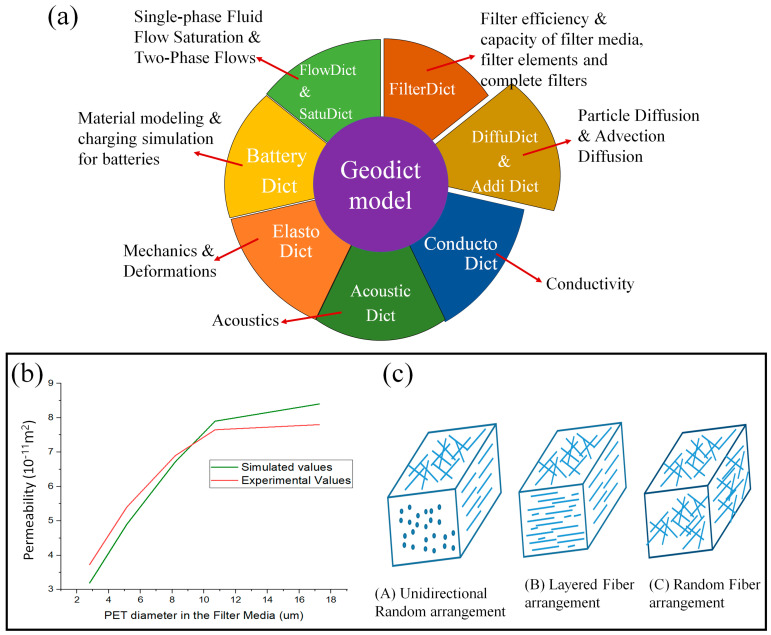
(**a**) Various models of GeoDict with its application [78], (**b**) Comparison of simulated and experimental data of filter media. Reproduce with permission from Ref. [93]. Copyrights 2019 Springer, and (**c**) three arrangements of fiber. Reproduce with permission from Ref. [94]. Copyrights 2008 Taylor & Francis Group, LLC.

GeoDict generates a 3D model using several inputs. A voxel-based algorithm in GeoDict constructs the unit cell, with each voxel in the domain either “empty” or “full”. When it comes to porous materials, the filled voxel represents the material’s solid portion, while the empty voxel shows its porous portion. Simulation of porous materials (made of powder) requires powder size, porosity, shrinkage, shape, and distribution, which can be varied according to the requirement. Geodict relies on Stokes flow to estimate permeability. Darcy’s law can be applied to the microscopic flow in the porous media [95]. Nada et al. [96] predict the thermal properties of the carbon GDL using a realistic 3-dimensional structure by ThermoDict. The dependence of thermal properties on porosity, fiber distribution, and compression was studied.

Further analysis was carried out on the relation between through-plane and in-plane thermal conductivity, and a significant difference was found. The whole simulation was carried out between 0.4% and 0.85% porosity. These results were compared with two experimental data and analyzed with a high degree of accuracy. Enis Tuncer et al. [97] also use GeoDict for the optimization of porosity with respect to mechanical and electrical properties. It was concluded that the best range for the required properties is between 0.4–0.55. Additionally, it was suggested that GeoDict is an excellent option for optimizing a variety of complicated properties that depend on a number of parameters. 

Geodict not only provides a result but can visualize the properties, which is helpful for in-depth research and understanding of how different properties rely on specific inputs. This feature makes GeoDict versatile and economical. P.-C. Gervais et al. [98] simulated filter media performance based on various parameters and compared it with experimental findings. Almost 97% accuracy was seen, as shown in Table 12. Where SV is the sub-volume, Ksim is the simulated permeability, Kexp is the experimental permeability, and MRE is the mean relative error. MRE is the difference between simulated and experimental results in percentage. P.-C. Gervais et al. [99] found the permeability with two different fiber sizes and suggested that the pressure drop and the filtration efficiency of the filter depend on the filter thickness and viscosity of the fluid. Their approach was to check these properties at various volume fractions and fiber diameter ratios denoted by α and R, respectively. The α was kept from 0.05–0.2, while R was 2–5. A comparison of GeoDict findings was made with two experimental data, and very high accuracy was observed at 0.138 < α < 0.193 and 2.6 < R < 4.1. 

An overview of the Geodict simulation of various porous material properties is presented. Although most of the simulation is about the fiber and ceramic material, very little attention is given to the titanium porous material consisting of powder. Therefore, it is recommended that with the same pattern, GeoDict can be applied to all metal-based porous materials, including GDL. The 2D and 3D models made of titanium powder generated by GeoDict can be seen in Figure 13. After shrinkage, the model is also shown, which is based on the real experimental calculation, and the shrinkage depends on the sintering temperature and demands of the final properties. By using this type of modeling, we can simulate multiple properties of the titanium GDL. This type of model can be generated with numerous inputs depending on accurate calculations such as the powder size, powder distribution, overall porosity, thickness, and shrinkage, to name a few. Researchers seek to optimize different properties, and numerous inputs are required for that. For this purpose, many simulations may be carried out using this tool in a short time and with lower expertise. 

First, the model is created in GeoDict using the inputs of the powder size and overall porosity, as illustrated in Figure 13a,b. The powder is then distributed using a particular iteration, shift distance, and random seed (Figure 13c,d). Followed by the sintered model with a defined amount of shrinkage based on the sintering temperature and shrinkage percentage (Figure 13e). After generating the model, several physical properties, including mechanical strength, pore composition, thermal property, and permeability, to name a few, can be determined. Figure 13f–h illustrates the physical views of pores, mechanical strength, and permeability. Their values rely on numerous variables, such as porosity, pores size, morphology, layer thickness, and material. This illustration provides an in-depth understanding of several properties and their governing agent. Hence, it is a proper alternative to the hectic, expensive, and lengthy experiment for optimizing an ideal prototype of porous material.

This review aims to understand various porous material manufacturing techniques and link them to the GeoDict simulation. It facilitates the reader to choose the best approach for creating porous material (GDL) based on their needs, available resources, and limitation. In addition, using GeoDict as an optimizing agent will save a significant amount of energy and time. As a result, high-performance titanium GDL will be efficiently designed and produced.

## 6. Conclusions and Recommendations

The high-power density, low operating temperature, and quick response of fuel cell and electrolyzer is the primary reason for their growth in the market. Electrolyzers store excess energy in the form of hydrogen, while fuel cells use stored hydrogen to reproduce electricity. For their high performance, well design parts with optimized properties are required. Porous GDL is a crucial component of both. The operation in the GDL is very complex due to its porous structure and simultaneous transport of charge, heat, and fluid. Highly efficient GDL include excellent permeability, high corrosion resistance, great thermal and electrical properties, and high mechanical strength. For all these properties, the material selection is essential, and then its fabrication with the optimized condition should be a significant concern. For material selection, titanium is the best candidate as compared to carbon and steel due to its high strength-to-weight ratio and corrosion resistance. In addition, porous titanium can play both functional and supportive roles.

Nevertheless, the excellent design structure is still challenging for good mechanical properties at high porosity. However, the structure pores design and alignment mainly depend on the fabrication method. Therefore, choosing an appropriate fabrication method according to the requirement should be the primary consideration because this is the fundamental component governing all the properties.

Tape casting is the most commercial method for fabricating GDL. This method has high control over the thickness of the GDL. The slurry composition and sintering temperature determine all porosity and pore morphology. Partial sintering controls the porosity, but the mechanical properties are very low at lower temperature sintering. For high porosity of around 90%, this method cannot be the right choice. Some improvements, such as employing space holders in tape casting and sintering at higher temperatures, are recommended to achieve high porosity with good mechanical strength. 3D printing is a versatile technique, and CAD design is the controlling factor of the whole material’s properties. The most popular method for printing Titanium GDL is powder bed fusion. 3D printing has a wide range of commands on the complex design of pores. However, in some cases, more accuracy is required for 3D printing in addition to being expensive and time-consuming.

Freeze casting is famous for aligned pores. The aligned pores vary with various types of freeze casting. The additive, freezing, and solidification mainly control the properties of the material. Further control is needed over the slurry formation, pore size, shape, and interconnectivity. PST is also a well-known commercial powder metallurgy technique for fabricating porous material. The porosity, pore size, shape, and other properties depend on the pore agent and its extraction method. In the future, this method needs to improve further control of the morphology, distribution of the pores, and contamination from the pore agent. This method also needs some improvement for a lower-thickness porous material. Cutting the thick material into thinner is one approach, but it will add some cost to the final products. Lithography could be employed to manufacture porous GDL. It uses a mask with a pattern that can be accurately transferred to the substrate by light exposure. The pores and porosity are controlled mainly by etching time and mask width. The future concern in this method is controlling the 3D structure of pores.

The second part of this review is the GeoDict simulation. Geodict is a versatile software with multiple modules that can simulate almost all properties of titanium GDL. New materials with various combination ratios can be predicted and optimized for future work. The effect of pores size, shape, binder, shrinkage, and others on the final properties can be analyzed. This tool will open up a new path for researchers studying GDL to optimize critical conditions without long and expensive experiments. Optimization of any properties needs many experiments. Sometimes, conducting experiments may be challenging when dealing with multiple properties simultaneously. For this problem, GeoDict is the most prominent solution. To sum up, this review aims to provide a concise overview of the various fabrication techniques for porous GDL and to connect them to software simulation to prevent high costs, energy, and time consumption.

## Figures and Tables

**Figure 1 materials-16-04515-f001:**
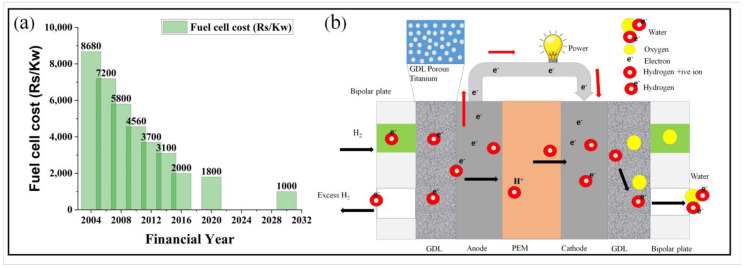
(**a**) decreasing price of fuel cells in the Indian market [8], and (**b**) schematic of PEM fuel cell.

**Figure 2 materials-16-04515-f002:**
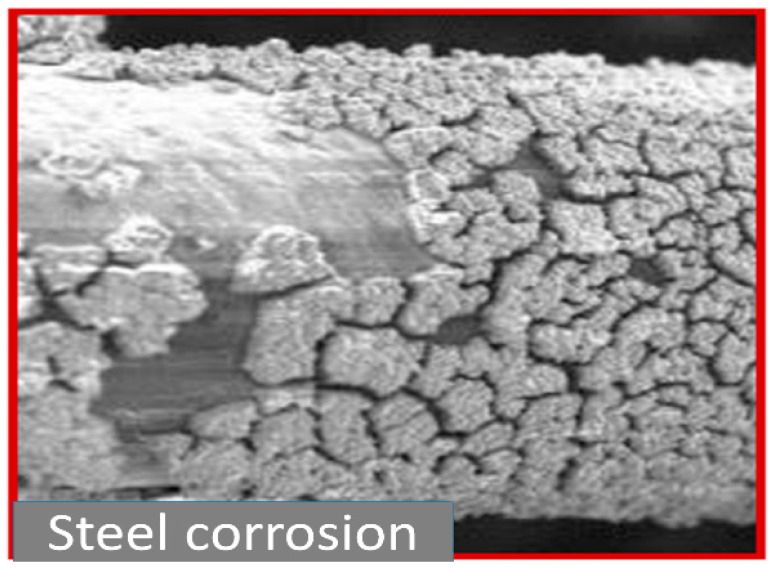
Carbon that comes out of steel cause corrosion, Reproduced with permission from [13]. Copyrights 2015 ECS.

**Figure 3 materials-16-04515-f003:**
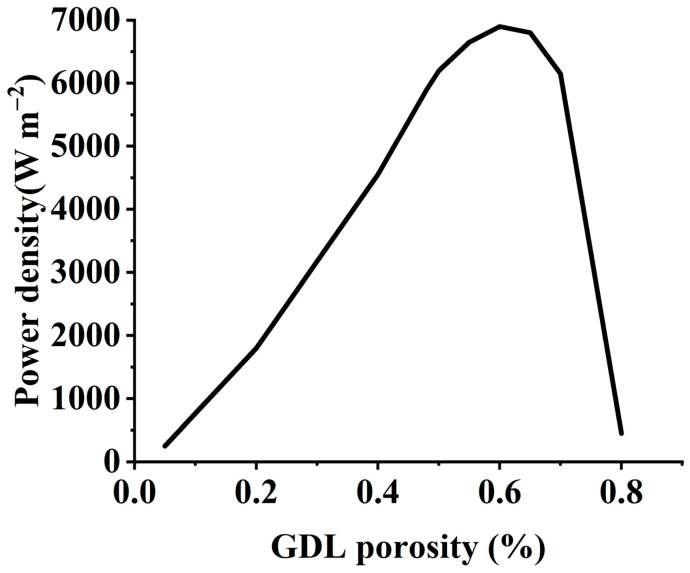
Porosity and power, reproduced with permission from [23]. Copyrights 2019 Elsevier.

**Figure 4 materials-16-04515-f004:**
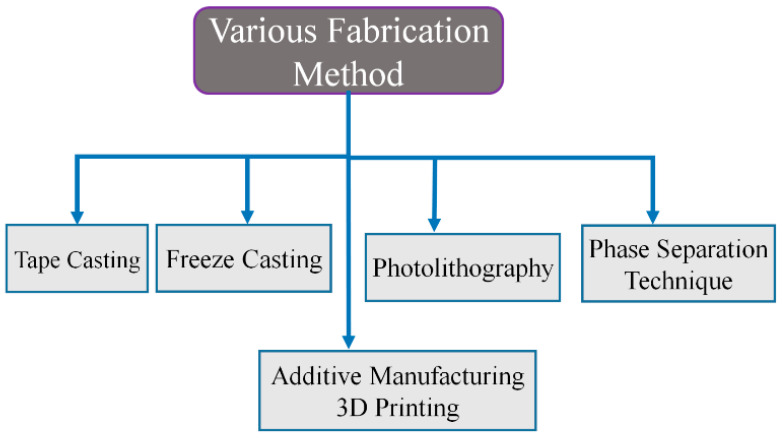
Fabrication methods of porous material.

**Figure 6 materials-16-04515-f006:**
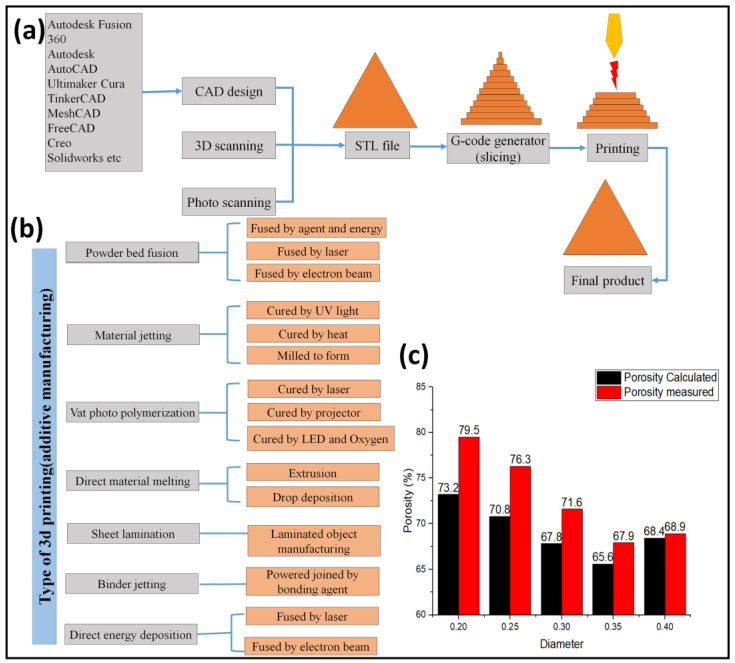
(**a**) 3D printing process, (**b**) various types of 3d printing [46], and (**c**) Comparisons of calculated and measured results. Reproduced with permission from [47]. Copyrights 2017 Elsevier.

**Figure 7 materials-16-04515-f007:**
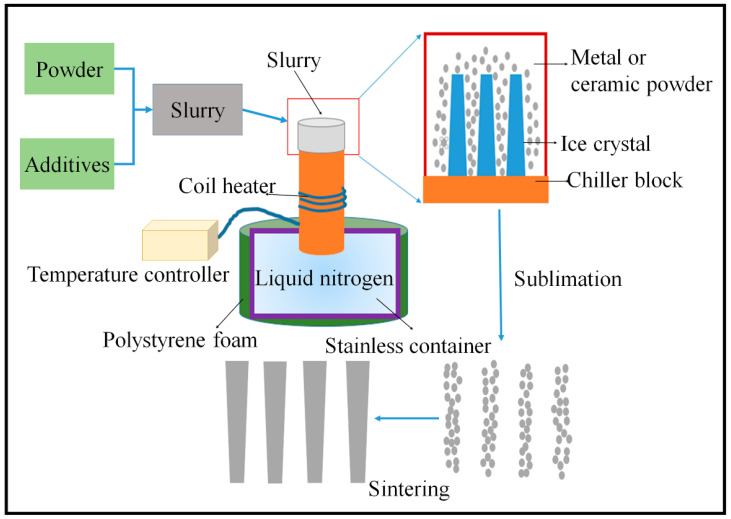
Various steps of freeze casting.

**Figure 8 materials-16-04515-f008:**
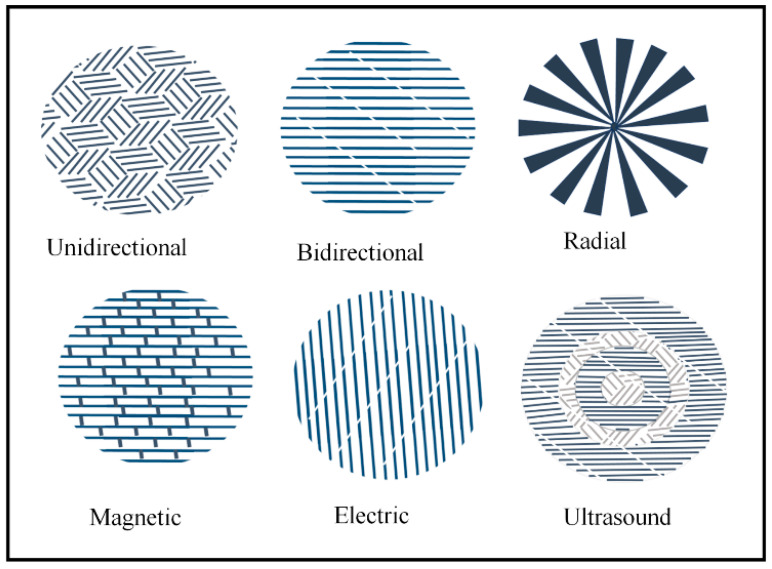
Types of freeze casting and the alignment of pores direction [52].

**Figure 9 materials-16-04515-f009:**
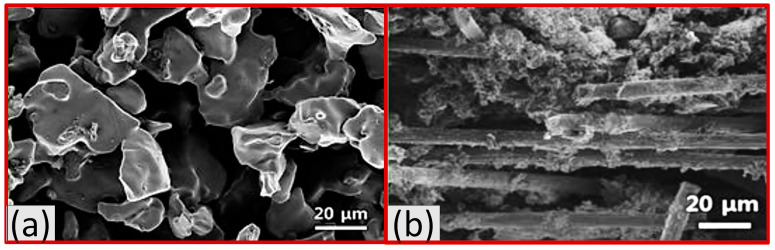
(**a**) Cross-section of SEM image of titanium foam GDL, and (**b**) Cross-section of SEM image of conventional GDL. Reproduce with permission from [54]. Copyrights 2014 ACS.

**Figure 10 materials-16-04515-f010:**
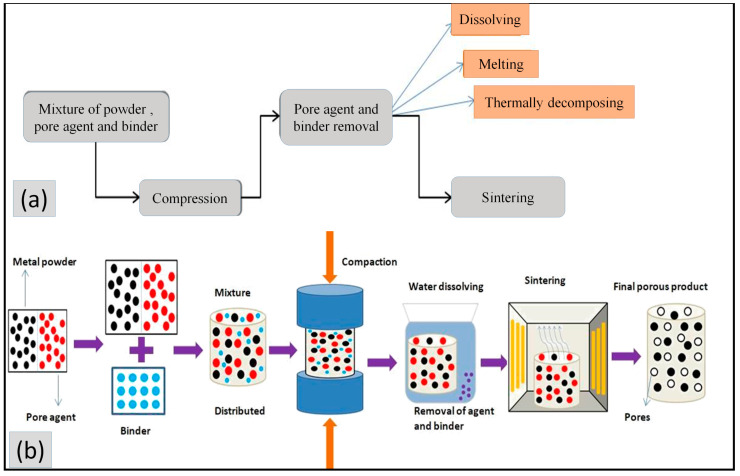
(**a**) Step in the phase separation technique, and (**b**) step in the sintering dissolution process.

**Figure 11 materials-16-04515-f011:**
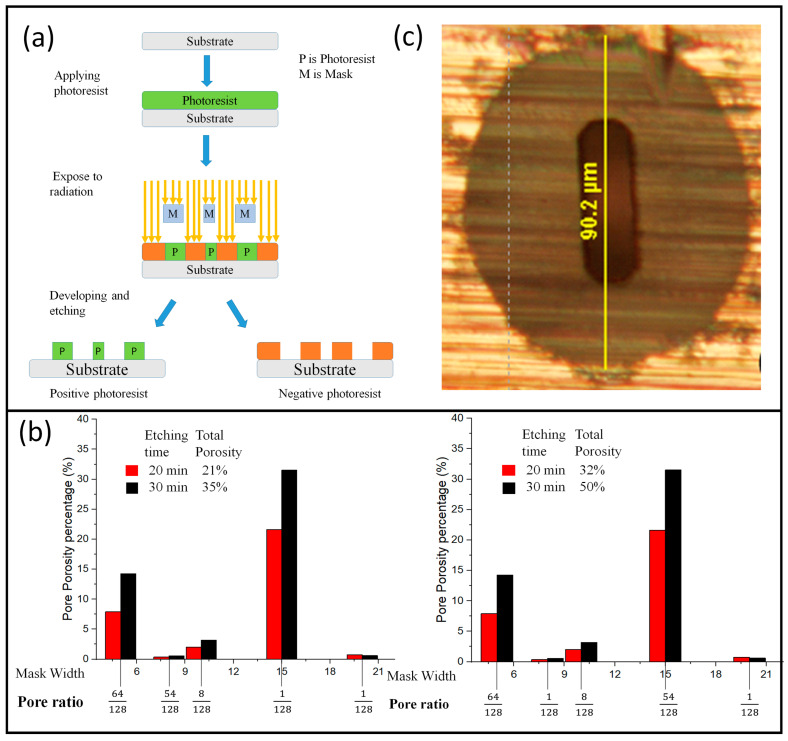
(**a**) Steps in the lithography technique, (**b**) porosity controlling by time and pore distribution, and (**c**) comparison of elliptical mask pore and dark brown final pore shape. Reproduced with permission from [2]. Copyrights, 2008 Elsevier.

**Figure 13 materials-16-04515-f013:**
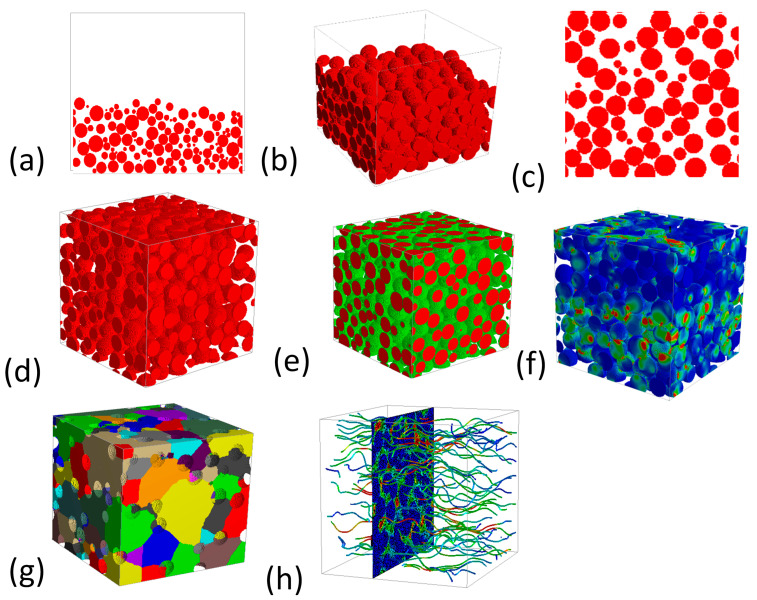
(**a**) 2D model view before distribution, (**b**) 3D model view before distribution, (**c**) 2D model view after distribution, (**d**) 3D model view after distribution, (**e**) 3D model view after shrinkage (sintering), (**f**) Schlieren 3D view demonstrating physical properties, (**g**) 3D view displaying various pores, and (**h**) Streamline view.

**Table 1 materials-16-04515-t001:** Various types of fuel cells based on electrolytes [6].

	Polymer Electrolyte Membrane Fuel Cell	Phosphoric Acid Fuel Cell	Alkaline Fuel Cell	Molten Carbonate Fuel Cell	Solid Oxide Fuel Cell
Electrolyte	Polymer membrane	Liquid H_3_PO_4_	OH^−^	CO_3_^2−^	O^2−^
Operating Temp (°C)	80	200	60–220	650	600–1000
Catalyst	Pt	Pt	Pt	Ni	Ceramic
Cell component	Carbon base	Carbon base	Carbon base	Stainless base	Ceramic base
Fuel	H_2_, CH_3_OH	H_2_	H_2_	H_2_, CH_4_	H_2_, CH_4_, CO
Efficiency	50–70	55	60–70	55	55–60

**Table 2 materials-16-04515-t002:** Properties of titanium and some important comparisons with carbon and steel. The electrical conductivity of steel is very low compared to carbon and titanium [16,17].

	Titanium	Carbon	Steel
Bulk modulus	110 Gpa	33 Gpa	160 Gpa
Mohs hardness	6	3	7–8
Electrical conductivity	2.5 × 10^6^ S/m	5.96 × 10^7^ S/m	1.43 × 10^−7^ S/m
Corrosion resistance	high	low	low
Density at 25 °C (g/cm^3^)	4.5	2.26	7.8

**Table 3 materials-16-04515-t003:** Additive for the making slurry [37,38,39,40,41].

Solvent	Binder	Dispersant	Plasticizer
Water	Polyvinyl alcohol	Phosphate esters	Polyethylene glycol
Methyl alcohol	Methylcellulose	Glyceryl trioleate	Diethyl oxalate
ethyl alcohol	Methy Methacrylate	Dolapix.	Tri ethylene glycol
Toluene	Methylethyl ketone	Dibutyl phthalate	Ethyl methacrylate
Acetone	Mowital.	Solseperse 20,000 g.	—

**Table 4 materials-16-04515-t004:** Properties of titanium GDL made by tape casting.

Material	Powder Size (μm)	Thickness (μm)	Porosity (%)	Sintering Temperature (**°C)**	Porous Structure	Ref
Ti	45	250–500	8.6–53.5	800–1200	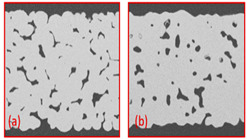	Reproduce with permission from Ref. [7]. Copyrights 2017 Elsevier
Ti-10Nb	0–100	350	1.5–7	(a) 1100 (b) 1150 (c) 1200 (d) 1300	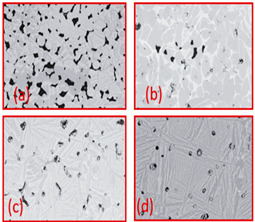	Reproduce with permission from Ref. [40]. Copyrights 2019 Elsevier.
Ti Spherical	-	231–551	6.6–55.6	(a) 800 (b) 900 (c) 1000 (d) 1200	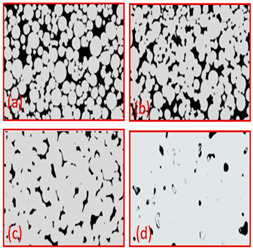	Reproduce with permission from Ref. [41]. Copyrights 2019 Wiley onine library.
Ti (HDH)	-	231–551	6.6–55.6	(a) 800 (b) 900 (c) 1000 (d) 1200	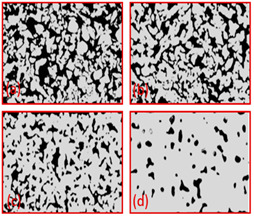	Reproduce with permission from Ref. [41]. Copyrights 2019 Wiley online library.
Ti64	-	370	36.2	1000	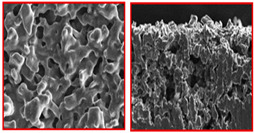	Reproduce with permission from Ref. [42]. Copyrights 2006 Elsevier

**Table 5 materials-16-04515-t005:** Dependence of pore morphology on various parameters of freeze casting.

Material	Factor	Effect on Pores Morphology	Ref
TiO_2_	Binder (a) 3 wt % PVA (cross-section parallel to ice growth) (a) 3 wt % PVA (cross-section Perpendicular to ice growth) (c) 6 wt % PVA (cross-section parallel to ice growth) (d) 6 wt % PVA (cross-section Perpendicular to ice growth)	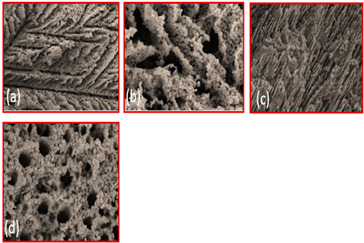	Reproduced with permission from Ref. [56]. Copyrights 2009 Elsevier
Alumina	Solid loading (a) 45% (b) 50% (c) 55% (d) 57.5% (e) 60%	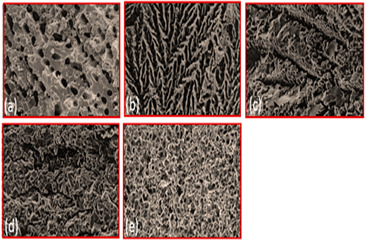	Reproduced with permission from Ref. [57]. Copyrights 2004 John Wiley
Hydroxyapatite	Solvent composition (a) Water (b) Water + 20% glycerol (c) Water + 60% glycerol	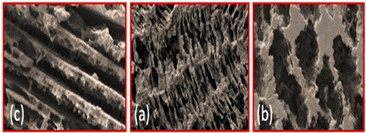	Reproduced with permission from Ref. [58]. Copyrights 2008 IOP Science
Hydroxyapatite/tri calcium phosphate (HA/TCP) ceramic	Freezing Temperature (a) 4 °C (b) 23 °C (c) 30 °C	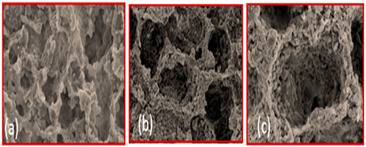	Reproduced with permission from Ref. [59]. Copyrights 2009 Elsevier
Al_2_O_3_	Freezing rate (a) 1 °C/min (b) 0.5 °C/min (c) 0.25 °C/min (d) 0.05 °C/min	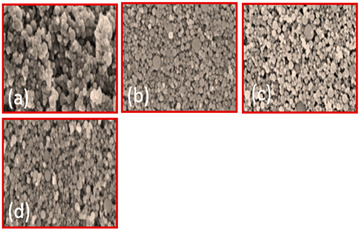	Reproduced with permission from Ref. [60]. Copyrights 2010 Springer

**Table 6 materials-16-04515-t006:** Porosity and pore size using various pore agents in the sintering dissolution method [1].

Pore Agent	Porosity Range (%)	Pore Size (**μm**)
NaCl	65–90	200–750
Carbimide	40–85	750–5000
Sacchrose	50–75	125–2000

**Table 7 materials-16-04515-t007:** Porosity and pore size using various pore agents in thermally stimulated decomposition [31].

Pore Agent	Porosity Range (%)	Pore Size (**μm**)
Polymethyl methacrylate	5–60	10–50
NH_4_HCO_3_	-	150–600
Saccharose	82	85–2000

**Table 8 materials-16-04515-t008:** Porosity and pore size using Mg and ice as a pore agent in thermally melted elimination [44].

Pore Agent	Porosity Range (%)	Pore Size (**μm**)
Magnesium	51–65	375
Ice	57–67	50–170

**Table 9 materials-16-04515-t009:** The detailed summary of various parameters of PST [31].

Method	Material	Pore Agent	Porosity	Pore Shape	Compressive Strength (MPa)
Sintering dissolution process	Titanium	Carbamide	44	Spherical Angular Needle-like	185 175 140
Thermally stimulated decomposition	Titanium Ti-15Mo	NH_4_HCO_3_	78 7.8 29.7 62.5	Spherical Spherical	35 1166 360 82
Thermally melted elimination	TiNi	Magnesium	49 58 64	Spherical	264.2–282.7 164–185.5 87.97–98.87

**Table 10 materials-16-04515-t010:** Comparison of the fabrication method.

Method	Porosity Controller	GDL Thickness Range	Basic Material
Tape casting	Sintering temp	100–2000 μm	Slurry
3D printing	Design of the 3D model	Micro size	Powder
Freeze casting	Ice	Micro and macro	Slurry
Phase separation technique	Pore agent	Micro and above	Only powder
Photolithography	Etching time and mask width	Micro to nano size	Photoresist filament

**Table 12 materials-16-04515-t012:** Summary of the flow simulation result compared with the experimental result [98].

SV	K_sim_ × 10^−11^ (m^2^)	MRE_exp/sim_ (%) K_exp_ = 3.36 × 10^−11^ m^2^
1	3.25	3.2
2	3.44	2.3
3	3.27	2.7
Mean	3.26	4.2

## Data Availability

Not applicable.

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
