# Peer review of "Porous Material (Titanium Gas Diffusion Layer) in Proton Exchange Membrane Fuel Cell/Electrolyzer: Fabrication Methods & GeoDict: A Critical Review"

_materials, 2023, doi:10.3390/ma16134515_

Round 1
Reviewer 1 Report
The article is well-written and gives a critical and comprehensive overview of the methods to fabricate porous Ti-based GDLs. I recommend publication in the present form. I found few minor errors to be corrected:
1. Abstract - PEMFC are not energy source (as written), they are energy conversion devices.
2. Introduction, p.2, line 61, authors should briefly explain what is Geodict.
3. Section 3, p.3, line 101, the authors state that "....is crucial for transport and water electrolysis", but before that they were discussing PEMFC.
4. Table 2 contains typo - electrical conductivity of steel.
5. p.5, line 179, thickness unit is wrong.
6. I recommend in general going through the manuscript and correcting typos.
Author Response
The authors cordially thank all the reviewers for taking up our manuscript titled " Porous material (Titanium gas diffusion layer) in proton exchange membrane Fuel Cell/ Electrolyzer: Fabrication methods & GeoDict: A critical review” for review. We thank the reviewers for the useful and insightful comments on our manuscript. We have tried to correct and improve our manuscript considering all the points below. The revised texts have been written in red color in the revised manuscript.

Reviewer 2 Report
Please see file in attachment.

Must be improved.
Author Response

(The authors gave the same response as above.)

Reviewer 3 Report
This review summarises the fabrication and simulation methods of the porous materials used as gas diffusion layers in PEMFCs. This work provides a focused topic and shows the related fabrication methods of Titanium porous materials. However, it still needs a more general description and comments about the PEMFC and the GDL. For example, the challenge of the fuel cell and the contribution of components to the performance of the fuel cell, such as the catalysts layer, gas diffusion layer, et al. This could provide context for the reader and make the review more accessible to those unfamiliar with the subject matter. Then, I think this manuscript might not be suitable for publication in “Materials”. More suggestions are listed in the following.
1. The list of the Types of Fuel cells in part 2 can be deleted because other fuel cells are not discussed in this review.
2. The authors should discuss the role of the GDL in the performance of PEMFCs by comparing that of each component, such as the catalysts layer, GDL and PEM, and provide more summary and comments.
3. It would be helpful to include a discussion on the impact of different GDL materials on the performance of PEMFCs. This could include a comparison of the advantages and disadvantages of different porous materials and how they affect the fuel cell's efficiency.
4. The relationship between the fabrication method and the GDL performance should be discussed and commented on. In this version, the preparation method is close to connecting the porous materials. When porous materials are applied to GDL, are there any other requirements that should be satisfied? And are these fabrication methods only used for the Ti materials?
5. More simulation methods should be summarised to understand how to design the ideal GDL. This could help readers understand the process of designing an effective GDL and provide insight into how to improve the performance of PEMFCs.
Author Response

(The authors gave the same response as above.)

Round 2
Reviewer 2 Report
Thanks to the authors for the corrections and responses to comments. I recommend checking the text for minor typos before publishing. Accept in present form.
Author Response
Response to Reviewer
The authors express their sincere gratitude to the reviewers for their valuable time and effort in reviewing our manuscript titled "Porous material (Titanium gas diffusion layer) in proton exchange membrane Fuel Cell/ Electrolyzer: Fabrication methods & GeoDict: A critical review”. We appreciate the reviewer' insightful comments and suggestions, which have greatly contributed to the enhancement of our manuscript.
Reviewer 3 Report
The author has addressed most of the issues in this article, but there are still some remaining corrections to be made. With the author's revisions, it is recommended to accept this article.
1. Please use the first author's name before "et al." and give the reference number after "et al.", not after the comma. Please cite the references at the correct locations and check the spelling of references in the text and the List of References.
2. Please review carefully and add the missing references, for example, on page 8, line 293: "Rauscher et al. reported that the viscosity of fine powder is higher than that of coarse powder, which requires more dispersion. "
3. In Table 2, the density is missing the unit. Please add it.
4. There are superscript and subscript errors in some units in the article, for example, on page 5, line 171, it should be "Titanium has a silvery white color and a density of 4.5 g/cm3, which is almost half that of steel with the same strength." Please carefully check and correct it.
5. Please verify and standardize the symbols used throughout the article, for example, "u" should be corrected to "μ" and "0C" should be corrected to "℃".
6. On page 13, line 432, the author's name should be corrected to "Hyelim Choi."

Author Response
Response to Reviewer
The authors express their sincere gratitude to the reviewers for their valuable time and effort in reviewing our manuscript titled "Porous material (Titanium gas diffusion layer) in proton exchange membrane Fuel Cell/ Electrolyzer: Fabrication methods & GeoDict: A critical review”. We appreciate the reviewer' insightful comments and suggestions, which have greatly contributed to the enhancement of our manuscript. The revised sections have been highlighted in red in the revised version.
The author has addressed most of the issues in this article, but there are still some remaining
corrections to be made. With the author's revisions, it is recommended to accept this article.
Comment 1: Please use the first author's name before "et al." and give the reference number after "et al.", not after the comma. Please cite the references at the correct locations and check the
spelling of references in the text and the List of References.
Our response to the comment: The authors would like to express their gratitude to the reviewer for their valuable comments. We have carefully reviewed the suggestions and have made the necessary corrections and one of these changes can be seen in the highlighted red text. We believe that these revisions have significantly improved the quality and clarity of the manuscript.
Modification to the manuscript:
Staurt et al. [19] evaluated three thicknesses (170um, 278um, and 534um) and found that as the thickness of the GDL reduced, the Ohmic loss and transport losses dropped, which led to an improvement in performance.
Comment 2: Please review carefully and add the missing references, for example, on page 8, line 293: "Rauscher et al. reported that the viscosity of fine powder is higher than that of coarse
powder, which requires more dispersion."
Our response to the comment: The authors sincerely thank the reviewer for their valuable comment. We would like to address the concern regarding the reference placement in the sentence. As per the suggestion, we have now corrected the reference placement and highlighted it in the red lines. Furthermore, we have carefully searched for any other missing references and made the necessary changes accordingly. We appreciate the reviewer's attention to detail and assistance in improving the accuracy of our manuscript.
Modification to the manuscript:
Section 4.1, Paragraph 6, sentence 1-2:
Rauscher et al. [35] reported that fine powder's viscosity is higher than coarse powder, which needs more dispersion.
Comment 3: In Table 2, the density is missing the unit. Please add it.
Our response to the comment: The authors sincerely appreciate the reviewer's valuable comment. We would like to inform that we have included the unit of density as recommended. This addition has enhanced the clarity and accuracy of our article. We are grateful for the reviewer's suggestion, which has contributed to the overall improvement of our work.
Modification to the manuscript:
Section 3.1.2, table 2:
|
|
Titanium |
Carbon |
Steel |
|
Bulk modulus |
110Gpa |
33Gpa |
160Gpa |
|
Mohs hardness |
6 |
3 |
7-8 |
|
Electrical conductivity |
2.5x106 S/m |
5.96x107 S/m |
1.43x10-7 S/m |
|
Corrosion resistance |
high |
low |
low |
|
Density at 25°C(g/cm3) |
4.5 |
2.26 |
7.8 |
Comment 4: There are superscript and subscript errors in some units in the article, for example, on page 5, line 171, it should be "Titanium has a silvery white color and a density of 4.5
g/cm3, which is almost half that of steel with the same strength." Please carefully check
and correct it.
Our response to the comment: The authors express their gratitude to the reviewer for their valuable comment. We have thoroughly reviewed the entire text and made necessary corrections to address all the superscript and subscript errors. As suggested, we have provided a sample of the revised text in the red line below. Thank you for bringing this to our attention and helping to improve the quality of our manuscript.
Modification to the manuscript:
Section 3.1.2, sentence 3:
Titanium has silvery white color and 4.5 g/cm3 density, which is almost half as compared to steel with the same amount of strength
Comment 5: Please verify and standardize the symbols used throughout the article, for example, "u" should be corrected to "μ" and "0C" should be corrected to "℃".
Our response to the comment: The authors thank the reviewer for the useful comment. The authors would like to respond that we have taken into account the suggestion regarding the unit correction, and as a result, we have made the necessary adjustments. One of the sample highlighting the corrected unit can be observed in the indicated paragraph below. Thank you for your insightful feedback, which has contributed to enhancing the accuracy of our manuscript.
Modification to the manuscript:
Section 3.1.2, paraghph 5, sentence 4:
Staurt et al. [19] evaluated three thicknesses (170μm, 278μm, and 534μm) and found that as the thickness of the GDL reduced, the Ohmic loss and transport losses dropped, which led to an improvement in performance.
Comment 6: On page 13, line 432, the author's name should be corrected to "Hyelim Choi.
Our response to the comment: The authors express their gratitude to the reviewer for providing valuable comments. The authors would like to respond by acknowledging that the complete name was indeed missing, and we have rectified this oversight. Additionally, we have conducted a thorough examination of the entire text to identify and rectify any similar mistakes. We greatly appreciate the corrections you have highlighted, as they have significantly improved the quality of our manuscript.
Modification to the manuscript:
Section 4.3, paragraph 6, sentence 1:
Hyelim Choi et al. [55] reported the fabrication of GDL through freeze casting. The first slurry was made by adding 0.28g PVA and 10 ml of distilled water to 11.25g of Ti powder.
